# Engineering Properties of Sweet Potato Starch for Industrial Applications by Biotechnological Techniques including Genome Editing

**DOI:** 10.3390/ijms22179533

**Published:** 2021-09-02

**Authors:** Ruiqing Lyu, Sulaiman Ahmed, Weijuan Fan, Jun Yang, Xiaoyun Wu, Wenzhi Zhou, Peng Zhang, Ling Yuan, Hongxia Wang

**Affiliations:** 1National Key Laboratory of Plant Molecular Genetics, CAS Center for Excellence in Molecular Plant Sciences, Chinese Academy of Sciences, Shanghai 200032, China; rlv222@uky.edu (R.L.); sulaiman@cemps.ac.cn (S.A.); wuxy@joyebio.com (X.W.); wzzhou@sibs.ac.cn (W.Z.); 2Department of Plant and Soil Sciences and Kentucky Tobacco Research and Development Center, University of Kentucky, Lexington, KY 40546, USA; lyuan3@uky.edu; 3Shanghai Key Laboratory of Plant Functional Genomics and Resources, Shanghai Chenshan Plant Science Research Center, Chinese Academy of Sciences, Shanghai 201602, China; wjfan@sibs.ac.cn (W.F.); jyang03@sibs.ac.cn (J.Y.); 4University of Chinese Academy of Sciences, Beijing 100049, China

**Keywords:** sweet potato, molecular genetics, starch metabolism, crop improvement, genome editing, biotechnology, CRISPR/Cas9

## Abstract

Sweet potato (*Ipomoea batatas*) is one of the largest food crops in the world. Due to its abundance of starch, sweet potato is a valuable ingredient in food derivatives, dietary supplements, and industrial raw materials. In addition, due to its ability to adapt to a wide range of harsh climate and soil conditions, sweet potato is a crop that copes well with the environmental stresses caused by climate change. However, due to the complexity of the sweet potato genome and the long breeding cycle, our ability to modify sweet potato starch is limited. In this review, we cover the recent development in sweet potato breeding, understanding of starch properties, and the progress in sweet potato genomics. We describe the applicational values of sweet potato starch in food, industrial products, and biofuel, in addition to the effects of starch properties in different industrial applications. We also explore the possibility of manipulating starch properties through biotechnological means, such as the CRISPR/Cas-based genome editing. The ability to target the genome with precision provides new opportunities for reducing breeding time, increasing yield, and optimizing the starch properties of sweet potatoes.

## 1. Introduction

Sweet potato (*Ipomoea batatas*) is one of the largest food crops in the world (Figure 1a). Although it originated from Central or South America, China is now the leading sweet potato producer in the world (Figure 1b). Sweet potato has become one of the most important food crops globally due to its superior stress tolerance and high yields [1,2]. Due to its high starch content and sustainable production, sweet potato provides raw materials for starch and starch-derived food, biofuel, and industrial products [3].

Starch biosynthesis requires four classes of core enzymes: ADP-glucose (Glc) pyrophosphorylase (AGPase), starch synthases (SSs; EC 2.4.1.21), starch branching enzymes (SBEs), and starch debranching enzymes (DBEs; EC 3.2.1.70) [4,5] (Figure 2). AGPase catalyzes the formation of ADP-Glc for the elongation of α−1,4-glucosidic chains [6]. SSs are categorized into five groups as granule-bound starch synthase (GBSS), SSI, SSII, SSIII, and SSIV [7]. In cereal crops, GBSSI is a key enzyme for amylose synthesis [4]. SSI, SSII and SSIII are responsible for the elongation of amylopectin [7]. SBE functions to generate 1,6-branch linkages, and acts as a key enzyme controlling starch granule structure and physicochemical properties [8]. The isoamylase and pullulanase activities of DBE play important roles in amylopectin synthesis [9].

Biochemical properties of starch vary among plant species, largely because of the ratio of amylose and amylopectin. The ability to modify starch properties for novel uses is of significant agricultural and industrial importance. Genetic engineering has emerged as a highly practical and cost-effective approach to alter the properties of starch, producing unique starch types for different industrial applications [10]. The starch biosynthetic pathway has been studied via genetic transformation of major crops, including rice [11], potato [12], and maize [13]. In sweet potato, the core genes involved in starch biosynthesis, including *AGPase* [14,15], *GBSSI* [16], *SSI* [7], *SSII* [17], and *SBEII* [16], have been investigated (Table 1).

In addition to the metabolic enzymes [26,27], transcription factors [28], stress responsive factors [29,30], and kinases [31] regulate the starch biosynthetic pathway. The involvement of a large number of genes reflects the complexity of starch molecules, amylose/amylopectin ratio, and granular characteristics. The complicated mechanisms governing starch biosynthesis also increase the difficulty of genetic engineering, which requires an in-depth understanding of genomics, epigenomics, and gene regulation at transcriptional, post-transcriptional, and post-translational levels. Sweet potato is a heterozygous hexaploid containing 90 chromosomes, 30 of which are derived from its diploid ancestor and the remaining 60 from its tetraploid ancestor. Although next-generation genomic sequencing technology has been established for more than a decade, a breakthrough in analysis of heterologous genomes has been achieved only recently. In addition, the first assembled sweet potato genome was published in 2017, and contains a high-resolution information of a half haplotype-resolved hexaploid genome of sweet potato. Due to the large natural variation of sweet potato plants and the complexity of its genome, the newly released sweet potato genome provides vital information for precise genetic manipulation by genome-editing technology [32]. Genome editing is a general term describing the technologies that allow specific changes of the DNA sequence in a genome, leading to alteration of genetic traits. Genome-editing technologies have been revolutionized by the recent development and application of the CRISPR/Cas system [33].

Progress in elucidating the sweet potato metabolic pathways and the structural and functional aspects of sweet potato starch has been described recently [21]. However, less coverage has been given to the correlation of the starch structure with functionality, or the modification of starch functionality for value-added products, especially through the application of genome-editing technology. Following a brief introduction of the starch biosynthetic pathway and the key enzymes involved, this review focuses on the regulation of carbon metabolism during storage root development, the CRISPR/Cas technology, and its applications in the modification of starch structure and quality.

## 2. Starch Biosynthesis in Sweet Potato and Other Plants

Starch biosynthesis is a complex and highly regulated process that requires coordinated activities of multiple enzymes, including AGPase, SS, SBE, and DBE (Figure 2). The starch biosynthetic enzymes and regulators control the structure and properties of starch [34]. The biosynthetic enzymes share high sequence similarity among plant species, especially in the functional domains, although the overall protein sequences may not be identical [35]. For example, the SS homologs share 60–80% similarity among maize, rice, and wheat. Nevertheless, the alignment of SS proteins from maize and barley with *Escherichia coli* glycogen synthase (EcGS) shows that multiple domains, required for binding of glucose, ADP, and maltopentaose, are conserved [35].

AGPase controls the reversible synthesis of ADP-Glc [6]. The IbAGPase activity in amyloplast is considered to be the main determinant of tuberous root formation [36]. SS functions to elongate linear glucan chains by catalyzing the transfer of the glucosyl unit of ADP-Glc to the nonreducing end of a glucan chain. To date, seven SS enzymes have been identified, namely, GBSS, SSI, SSII, SSIII, SSIV, SSV, and a newly identified SSVI in cassava (*Manihot esculenta)* [7,37]. SS can be broadly divided into two groups; the first group (GBSS) is primarily involved in amylose synthesis, and the second (the remaining SS enzymes) is confined to amylopectin production [38]. GBSS transfers the glucosyl residues from ADP-Glc to its glucan substrate to generate the long glucan chains. It also acts on the existing side chains of amylopectin and contributes to the formation of long chain amylopectin.

SS family proteins are soluble in amyloplasts of the chloroplast stroma. SSI preferentially elongates newly placed branches to a length around 8–10 Glc units. SSII further elongates these chains to around 13–18 Glc units. In maize, SSII forms a trimeric complex with SSI and SBE1I in maize amyloplast stroma, regulating glucan branching [39]. SSIII is proposed to synthesize long cluster-spanning amylopectin chains and is conserved in both monocots and dicots [40,41]. SSIV coordinates granule formation that leads to the flattened, discoid shape of leaf starch granules during leaf expansion [42]. SSV is homologous to SSIV but lacks the C-terminal GT1 subdomain and is conserved in all green plants. Rather than directly regulating starch granule initiation, SSV affects other network components to promote the initiation of starch granule [43]. SSVI only exists in dicots. Knockdown of SSVI in cassava retards plant development and increases the average granule size [37]. The cassava SSVI protein potentially influences the activities of AGPase, GBSS, and ISA, by forming a protein complex with the key starch biosynthetic enzymes (SSI, SSVI, SBEI, SBEII, ISAI, ISAII, and GBSSI) [37].

SBE catalyzes the formation of branch points by cleaving the α-1,4 linkage in polyglucans and reattaching the chain via β-1,6-glucan linkage [44]. Plants possess two classes of SBE, i.e., SBEI and SBEII, based on biochemical and physicochemical properties. Transgenic sweet potato plants with repressed *SBEII* (*IbSBEII*) by RNAi accumulate higher amylose than the wildtype plants (up to 25% compared to 10% in the control) [20].

DBEs (α-1,6-glucanohydrolases) cleave branch points and determine the structure of amylopectin [45]. The two classes of DBE, isoamylase (ISA) and pullulanase (PUL), directly hydrolyze the β-1,6-glucosic linkages of polyglucans. ISA mainly debranches phytoglycogen and amylopectin, whereas PUL acts upon pullulan and amylopectin, but not phytoglycogen. Few studies have been conducted on DBEs in sweet potato. At least two copies of *IbIsa1* are present in the sweet potato genome [45]. *IbIsa1* strongly expresses in the tuberous root. IbIsa1 likely works in concert with the AGPase large subunit, GBSSI, and SBEII during the initial stage of starch granule formation.

## 3. Functionality and Regulation of Biosynthesis of Starch

Starch contents vary from 4.5 to 31.8% in fresh storage roots of different sweet potato cultivars [17,44,46,47,48,49]. Sweet potato starch is a mixture of linear or slightly branched amylose and highly branched amylopectin, usually present at 20–30% and 70–80%, respectively [47]. Amylose is both a diluent and an inhibitor of swelling and is required for starch retrogradation. The chain length and the ratio of amylopectin/amylose determine the functional properties of starch [50,51]. Long-chain starches contribute to higher viscosity and stability of the starch gel compared with short-chain starches [52].

Amylose and amylopectin of different lengths form supramolecular structures with different length scale and molecular weights. There are four types of supramolecular structures, ranging from the smallest to the largest by size: (i) crystalline and amorphous lamellae (4–6 nm); (ii) amylopectin clusters (~9 nm); (iii) semi-crystalline and amorphous rings (120–400 nm); and (iv) granules (0.5–100 µm) [53,54,55,56]. Because starch forms a semi-crystalline granule with different supramolecular structures, sweet potato starch is categorized as A-type with ~34% of relative crystallinity based on X-ray diffraction patterns [57]. The A-type starch is composed of double helices of crystalline lamellae and packed into the polymorphous forms with monoclinic packing [58]. In addition, amylopectin chains, with a degree of polymerization (DP) between 10 and 24, are the dominant source forming the supramolecular structures in starch granules. By comparison, the shorter chains, with DP <10, tend to form an imperfect starch structure [58].

Starch content, chemical properties, amylopectin/amylase ratio, and supramolecular structures determine the size of starch granules [59,60]. The distribution of starch granules is a critical characteristic and a vital factor for the quality of sweet potato-derived final products. Sweet potato starch granules are found in various shapes and sizes, from round to polygonal, oval, and semi-oval [61]. The formation of different granules is not only related to growth and agronomic management, but also to the expression of genes, especially those associated with starch biosynthesis [7,22,24,31].

Starch metabolism is closely related to storage root development and correlates with starch degradation by β-amylase, one of the major enzymes in sweet potato storage roots [28,62]. RNA sequencing and microarray data show that genes related to starch and sugar metabolism express differentially between sweet potato fibrous roots and storage roots, indicating the key role of carbon metabolism during storage root development [63].

Additionally, transcription factors (TFs) regulate carbohydrate metabolism during sweet potato storage root development. The sweet potato Dof-zinc finger TF *SRF1* is highly expressed in storage roots [62]. Transgenic sweet potato overexpressing *SRF1* contains higher starch and lower monosaccharide content. The reduced expression of vacuole invertase (*IbBfruct2*) in the transgenic plants suggests that SRF1 represses the enzyme to regulate carbohydrate metabolism [62]. Moreover, the cassava sucrose synthase (MeSus1), an important gene for starch biosynthesis in the storage root, is negatively regulated by an ethylene responsive factor, MeERF72 [64]. In cassava, abscisic acid (ABA) is a potential inducer of *SBE* through the phosphorylation signal cascade [65], suggesting that carbohydrate accumulation during storage root development is regulated by phytohormones, although more experimental evidence is needed to establish such a connection.

Proteomic studies provide a list of proteins that function in the regulation of starch and sugar metabolism during cassava storage root tuberization [66]. Carbon assimilation is tightly connected with nitrogen metabolism. The treatment with calcium nitrate [Ca (NO_3_)_2_] induces the sugar responsive kinase gene *IbSnRK1*. Overexpression of *IbSnRK1* in sweet potato not only enhances photosynthesis and carbohydrate accumulation, but also increases nitrogen uptake efficiency [67]. Fertilization also affects carbohydrate distribution. High nitrogen application leads to reduced root yield in both nitrogen-tolerant and nitrogen-susceptible sweet potato varieties. Nitrogen-tolerant varieties show more carbon allocation in tuberous roots under no-nitrogen conditions compared with nitrogen-susceptible varieties [68]. These results indicate that nitrogen regulates carbon flux through mechanisms that are yet to be determined, and a balance between nitrogen and carbon metabolism is required during sweet potato root development. Overexpression of the maize *Lc* (leaf color) gene, involved in flavonoid biosynthesis, suppresses sweet potato storage root expansion and results in increased lignin synthesis and decreased starch accumulation in storage roots at the initiation stage [69]. A natural plant growth regulator, calonyctin, has been found to accelerate sucrose and starch synthesis during storage root formation [70], although the mode of action remains to be discovered. The increasing availability of genomic information for sweet potato [32] will be key to unraveling the regulation of carbon flux by hormone crosstalk, post-transcriptional regulation, and signal transduction.

The genetic regulation of starch metabolism in sweet potatoes and other crops determines the starch properties by affecting the amylose/amylopectin ratio, chain length, and distribution of amylose and amylopectin [51]. Because starch functionality often affects the specific end use, genetic modifications of starch metabolism-associated genes potentially enable the production of specialized starch for a specific industrial application.

### 3.1. Pasting and Gelatinization Properties

Pasting properties are characterized by the viscosity developed from a programmed heating and cooling cycle with a constant shearing force [71,72]. Pasting behavior involves granular swelling, leaching of amylose from starch granules, and the subsequent solubilization to form a starch paste [61]. The common parameters for pasting properties are peak viscosity (PV), breakdown (BD), setback (SB), and pasting temperature (Pte). The starch pasting properties of transgenic sweet potatoes compared to WT have been established using various methods. Gelatinization, the commonly known characteristic of starch, reflects the inflicted changes in granule swelling, crystalline melting, and amylose leaching [73]. During gelatinization, starch granules absorb water and swell to melt the internal crystalline structures, leading to the rupture of granules and disordering of the chain organization [74]. A significant genetic diversity associated with gelatinization properties has been observed for sweet potato starch [75]. Environmental factors, such as soil temperature, apparently influence gelatinization because higher soil temperature results in higher gelatinization temperature and melting enthalpy [46,76]. Increased soil temperature also leads to reduction in short chain amylopectin with DP 6–7 [46]. Growth temperature may also affect the crystal surface energy in the granules and crystalline lamellae thickness [76]. How the environmental factors are perceived, in terms of triggering signal transduction that alters gene expression in sweet potato, remains largely unknown.

### 3.2. Amylose Content and Amylose/Amylopectin Ratio

Amylose content is one of the most important parameters to be considered for starch properties and industrial applications [77,78]. During the heating process, amylose leaches rapidly from granules and aggregates to form amylose junction zones through hydrogen bonding [79] Amylose re-association is believed to be responsible for SB and short-term retrogradation [80]. Amylose content influences change in the starch properties more than other starch characteristics [81]. High amylose starch exhibits a decrease in BD and an increase in SB values because more amylose is leached from the granules [22]. The wide range of amylose content in sweet potato starch thus provides versatile applicability.

The amylose content reaches 65.5% in a transgenic sweet potato [22]. Amylose/amylopectin ratio is critical in determining the starch physicochemical and functional properties [73]. The amylose/amylopectin ratio is regulated by starch biosynthetic enzymes [38] and genetic variability [82]. The amylose/amylopectin ratios of 507 sweet potato germplasms range from 0.247 to 0.429 [44]. Changing the amylose/amylopectin ratio by altering the amylose content is a practical approach for improving the performance of starch derivative products.

### 3.3. Starch Granule Size

The sizes and distributions of starch granules are important factors in determining the physicochemical properties of starch for different applications [22]. The starch granule size from different plant sources varies from 1 µm to greater than 100 µm. Small granule starch (diameter < 10 μm) has higher solubility and water absorption capacity, and is thus easier to digest chemically and enzymatically in industrial applications. Sweet potato starch granules are generally round, oval, or polygonal in shape, ranging from 5 to 90 µm in diameter, with an average size of 19 µm (Figure 3).

The transgenic waxy sweet potato with high amylose have larger starch granule sizes (70 and 90 µm) compared with the wild type (WT) [22]. Reduction in amylose content reduces the average diameter of the starch granules; that is, higher amylose content tends to produce larger granules [10,22]. Additionally, starch granules from purple-fleshed sweet potato are smaller in size than those from white- and orange-fleshed sweet potato [83].

### 3.4. Chain Length Distributions (CLDs)

CLDs dictate the primary structure of starch [78]. CLDs and molecular sizes of starch are influenced by the degree of polymerization (DP) of amylose and amylopectin. Five common fractions (%) are used for discriminating CLDs of amylopectin, including DP 6–12 (fa), DP 13–24 (fb1), DP 25–36 (fb2), and DP ≥ 37 (fb3). In sweet potato, the proportions of each fraction of chains in various cultivars have been reported. High-performance anion-exchange chromatography with pulsed amperometric detection (HPAEC-PAD) has been used to detect significant differences in the chain fractions between WT and transgenic waxy or high-amylose sweet potato [22]. The starches from sweet potato (with average chain length (ACL 20.4–24.7) and potato (ACL 20.5) have a similar range of CLD values, which are higher than those of cassava (ACL 19.3) and maize (ACL 18.9) [85].

Based on size-exclusion chromatography (SEC), structures of amylose and amylopectin of sweet potato starch are at DP ≥ 100 and DP < 100, respectively [78]. Two peaks were generated for each of amylose (DP 100–700 and DP 700–20,000) and amylopectin (DP ~17 and DP ~41) from debranched sweet potato starch [78]. In addition to SEC, fluorophore-assisted carbohydrate electrophoresis has been used for CLD analysis of small chains (DP < 100) of amylopectin [78]. The highest CLD of sweet potato was at DP 13–24, ranging from 45.5% to 59.8%. Moreover, gelatinization properties are related to CLD, structural glucan chains, proportion of amylopectin fractions, and glucan compositions [86]. The higher CLDs of short amylopectin may contribute to the lower gelatinization temperatures. One benefit of lower gelatinization temperature is the reduction of energy consumption and CO_2_ emission in bioethanol production [87]. Recently, nanoscale chains in starch lamellae in transgenic sweet potato have been detected using the small-angle-X-ray-scattering (SAXS) technique [23]. The high-amylose starch appears to contain a greater quantity of the newly identified semicrystalline lamellae (Type II; 0.040 Å-1 thickness) than the waxy starch. In contrast, WT sweet potato has only Type I lamellae (0.065 Å-1). Compared to the Type I lamellae, the Type II lamellae shows increased average thickness, in addition to thickened amorphous and crystalline components. By downregulating the expression of SBE or GBSSI, the level of Type II lamellae increases in the transgenic sweet potato.

### 3.5. Starch Phosphorylation

Starch phosphorylation is a naturally occurring chemical modification; however, its physiological function is not known. Phosphorylated creamy starch, such as that from potato tubers, is easy to hydrate, producing a clear and sticky paste. Similar functionalities can be achieved though industrial chemical treatments. Detailed descriptions of starch phosphorylation have been nicely reviewed [88,89,90,91]. Here, we summarize recent progress regarding the mechanism and genetic manipulation of starch phosphorylation.

Thus far, two kinases, the glucan, water dikinase (GWD1) [92,93] and phosphoglucan, water dikinase (GWD3 or PWD) [94], have been characterized for starch phosphorylation. GWD1 selectively catalyzes the addition of phosphate monoesters at the C-6 position. GWD3 activity, which depends on GWD1 action, is required for the addition at the C-3 position [92]. Modulation of *GWD1* expression altered the starch phosphate content in potato tubers, indicating a direct link between *GWD1* expression and starch phosphorylation levels [95]. Although the starch content remained unchanged, the content of amylose showed a negative correlation with GWD1 expression. This is likely because the expression of *SP*, *SSII*, *SSIII*, and *SBEII* was affected by the starch phosphate content [95]. These results suggest that starch phosphorylation affects starch biosynthesis. A similar function for GWD1 was also observed in cassava [96]; however, the cassava GWD1 affected the transient starch morphogenesis. Two phosphoglucan phosphatases, STARCH EXCESS 4 (SEX4) and Like-SEX Four 2 (LSF2), have been reported for amylopectin dephosphorylation. SEX4 releases phosphate from C3 and C6 positions of Glc, with a preference for C6. LSF2 specifically releases C3-bound phosphates [97,98,99]. The relationship between starch properties and starch phosphorylation suggests that modulating starch properties to meet industrial needs can be achieved by controlling starch phosphorylation. This has been successfully demonstrated in cassava via RNAi [100].

## 4. Value-Added Products from Sweet Potato Starch

Sweet potato starch is used in various food and industrial applications. Sweet potato starch is particularly valued as an important ingredient in the manufacturing of starch-based food products, such as noodles, vermicelli, jellies, steamed bread, cakes, alcoholic beverages, soup, flavoring agents, sweeteners, and other consumables [61]. Sweet potato starch is suitable for producing resistant starch, which helps to reduce the postprandial blood glucose level and reduces the risk of obesity and diabetes [101]. Resistant starch is considered to be an important value-added starch product with increasing market demand. A promising area for sweet potato starch is in the renewable replacement of petroleum feedstock, e.g., biofuels and biodegradable plastics [102]. Starch-based film shows significant potential to replace conventional plastic films based on its biodegradability, relative abundance, chemical inertness, and resistance to chemical or enzymatic degradation [103,104]. However, the existing starch-based film has shown poor mechanical and barrier properties, which are caused by different factors, such as amylose/amylopectin ratio, granular morphology, granule size, and granule size distribution [105]. Changes in starch properties to improve the quality of starch films can be achieved chemically, enzymatically, or physically [106]. The amylose/amylopectin ratio affects the tensile strength and elastic modulus [107]. An environmentally greener approach is perhaps through genetic manipulation that produces sweet potato varieties with desired starch properties (e.g., higher amylose/amylopectin ratio).

One promising attempt was made to fill a thermoplastic starch matrix with nanofillers, because the nanoscale particles of starch showed different crystallization kinetics, resulting in varied crystalline morphology and size [108]. Compared with potato starch, normal sweet potato starch is not ideal for making films due to its poor mechanical and water vapor barrier properties [109]. Nonetheless, potassium sorbate- and chitosan-incorporated sweet potato starch films have antimicrobial activities. Antimicrobial packaging is another promising area of the food industry that has received growing attention [110]. Starch-based nanoparticles display unique properties, such as controllable release, improved water solubility, bioavailability, and improved delivery of active ingredients in foods and within the human body [111,112]. In addition to enhancing the properties of starch films, starch-based nanoparticles can be used to produce high value-added products, such as the biodegradable carriers for drug delivery [113].

Starch-based nanoparticles are conventionally produced by acid hydrolysis or high-pressure homogenization, although enzymatic hydrolysis is more effective and environmentally friendly. Starch-based nanoparticles with desired sizes can be produced using amylases or/and debranching enzymes to digest the α-1,4-glucosidic or α-1,6-glucosidic bonds in starch. Enzymatic hydrolysis is likely to yield homogeneous nanoparticles when the starch contains more A and B type short chains with an average DP of 14–18 [112]. Sun et al. reported a method of producing starch granules of 60–120 nm in size from maize waxy starch that has 99% amylopectin [114]. Waxy starch that lacks amylose, thus resisting retrogradation, is more suitable for food and polymer applications, in addition to the production of starch-based nanoparticles [115]. New functionalities are desired for sweet potato starch, which are difficult to achieve through traditional crop breeding due to its long duration and the complex genetic background. Recent advancement in biotechnology, particularly genome-editing technology [16], offers a new approach to generate novel functionality for sweet potato starch.

## 5. The Promise and Challenge of Genome-Editing Sweet Potato

The genome-editing technology CRISPR/Cas has emerged to be the tool of choice to manipulate genomes (Figure 4). After the first genome editing by CRIPSR/Cas9 published in 2012 [116], the technology has been widely used in prokaryotes and eucaryotes, including in plant studies [117,118]. Using genetic methods, the T-DNA that carries the sgRNA and CRISPR/Cas can be removed after the mutation is achieved, creating a T-DNA-free progeny (Figure 4). In the United States, the regulatory agencies generally recognize the T-DNA-free progeny as non-genetically modified (non-GM), based on the absence of T-DNA and the view that similar mutations can result from other conventional breeding techniques, such as chemical or UV-induced mutation. However, the European Union and some other countries do not view such plants as non-GM [119].

In addition, genome editing has proven to be efficient for multiple unlinked-loci mutagenesis and plants with complex genomes, e.g., the hexaploid sweet potato with 90 chromosomes, in addition to the polyploid wheat and potato [125,126]. By mutating eight potato *SBE* alleles using CRIPSR/Cas9 technology, a novel potato starch with no detectable branches was produced [127]. The first successful demonstration of CRISPR/Cas9 technology in sweet potato targeted two starch biosynthetic genes, *IbGBSSI* and *IbSBEII*, in the starch-type cultivar Xushu22 and the carotenoid-rich cultivar Taizhong6. The mutation efficiency was 62–92% with multi-allelic mutations in both cultivars [16], providing support for the effective use of CRISPR/Cas9 technology to improve starch qualities in sweet potato and to advance the breeding of polyploid root crops.

The use of well-established biotechnological tools to improve tuberous crops is effective. For example, silencing the vacuolar invertase inhibitor gene by RNAi improves the resistance to cold-induced sweetening of potato without affecting tuber quality when stored in cold conditions [128]. However, such an approach usually requires the presence of transgenes in the engineered plants, an issue that the food industry often tries to avoid. The application of CRISPR/Cas technology, in some cases, circumvents this issue [129]. The industry commonly modifies starch properties through physical and chemical methods, but the approaches are neither economical nor environmentally friendly. Enzyme-based modification of starch property improves the processes [130]; however, it is cost and energy intensive. Genome-editing technology potentially allows the modification of starch property in planta, thus by-passing the expensive industrial process.

Although genome editing is superior in many aspects compared to other biotechnological platforms or chemical mutagenesis in conventional crop breeding, possible off-targeting (mutating a non-targeted genomic site), due to low sgRNA specificity and the lack of genomic information for some plant species, can hinder its wide applications. Nevertheless, the second generation of CRISPR technology has significantly reduced off-targeting [121,131]. The first draft genome of sweet potato, released in 2015, is low resolution [132]. An updated genome, released two years later, significantly refined the sweet potato genomic information [32] and is pivotal in guiding successful sweet potato genome editing. Nonetheless, the sweet potato genome still remains difficult to annotate, because bioinformatic tools for analyzing polyploid genomes are not well developed [132]. Gene transformation and regeneration of sweet potato continue to be an engineering bottleneck [133]. Not all varieties can be sufficiently transformed using the Agrobacterium-mediated methods. However, it is clear that the newly developed CRISPR/Cas system [33,122,134], advanced bioinformatic analysis, and a novel delivery method [135] will mitigate these issues.

## 6. Discussion

Starch from sweet potato possesses certain unique chemical and physical characteristics compared to that from other sources. The existence of many germplasms with varied starch contents and characteristics establishes a genetic base for novel sweet potato crops that are suitable for many food and industrial applications. CRISPR/Cas-based genome editing has been successfully demonstrated in sweet potato and will expand the diversity of natural starch without the need of physical and chemical treatments.

Genetic manipulation of starch properties has advantages over physical or chemical methods, which are often associated with environmental pollution, expensive equipment, and special design and optimization in production [134,136]. For example, high hydrostatic pressure processing (HHP) is an effective method to physically modify starch properties, especially for gelatinization [136]. HHP can be performed at room temperature, but requires high pressure up to 100–1000 MPa, which may disrupt starch microstructures (reviewed in [136]). Other promising physical methods, including gamma radiation [137], microwaves [138], and cold plasma [139], show limitations. For instance, it is difficult to precisely control the experimental parameters using microwaves, thus affecting the quality of end products [136].

In contrast, genetic engineering methods are more environmental and cost friendly. Genetic manipulation of genes, using gene overexpression, RNAi, and genome editing, has been shown to be effective in the production of sweet potato starch with altered functionality. However, difficulties remain in genetic engineering of sweet potato. Starch properties are generally controlled by multiple genes, and simultaneously targeting multiple genes is an engineering challenge. Due to the complexity and insufficient annotation of the sweet potato genome, it is currently difficult to manipulate multiple loci by upregulation, knockout, or their combination. Due to the rapid advancement in gene sequencing and genomic analyses, these issues will be overcome in the near future. In addition, the generation of transgenic plants is time consuming, especially for crop plants, such as sweet potato, that are known to be difficult to regenerate in tissue culture. Nevertheless, we are now able to overexpress genes, knockdown gene expression, and knockout genes in sweet potato. These abilities allow us to generate libraries of sweet potato lines with varied starch properties. These libraries will be the basis for designer starch production through crossing or secondary gene transformation.

CRISPR/Cas genome editing technology has fundamentally changed agricultural breeding. Since the invention of the first CRISRP/Cas9 system, novel tools, such as CRISPR/Cas-mediated chromosome engineering, and Cas13 a, have been developed, and the development of more genome-editing tools is underway [140,141]. These tools will undoubtedly be applicable to the modification of sweet potato starch. Nonetheless, our continuing understanding of the biochemical and genetic determinants of sweet potato starch biosynthesis, combined with advances in transgenic manipulation, offer potentially accelerated approaches to achieve our goals. The complete genome sequence allows the identification and cataloging of potential genes that determine sweet potato starch biosynthesis and functionality. The newly updated genomic information provides targets for genome editing to improve starch yield, quality, and functionality, in addition to desirable agronomic characteristics of sweet potato.

## 7. Conclusions

The ability to naturally modify sweet potato starch significantly enhances the economic value of the crop. In this review, we provide an overview of sweet potato starch metabolism, describe the direct and indirect links between starch properties and genes involved in sweet potato starch metabolism, and associate different starch properties with final industrial products. The connections among genes, starch properties, and industrial products are the basis for modification and improvement of sweet potato starch through techniques such as genome editing. Gene knockout by CRISPR/Cas technology has another advantage compared to many other gene-silencing technologies, namely, the T-DNA carrying the CRISPR/Cas can be removed from the genome through segregation, thus significantly reducing the burdens in the governmental regulatory process. In summary, the maturation of the enabling technologies described here has opened the door to a new era of precision molecular breeding of sweet potato [142,143].

## Figures and Tables

**Figure 1 ijms-22-09533-f001:**
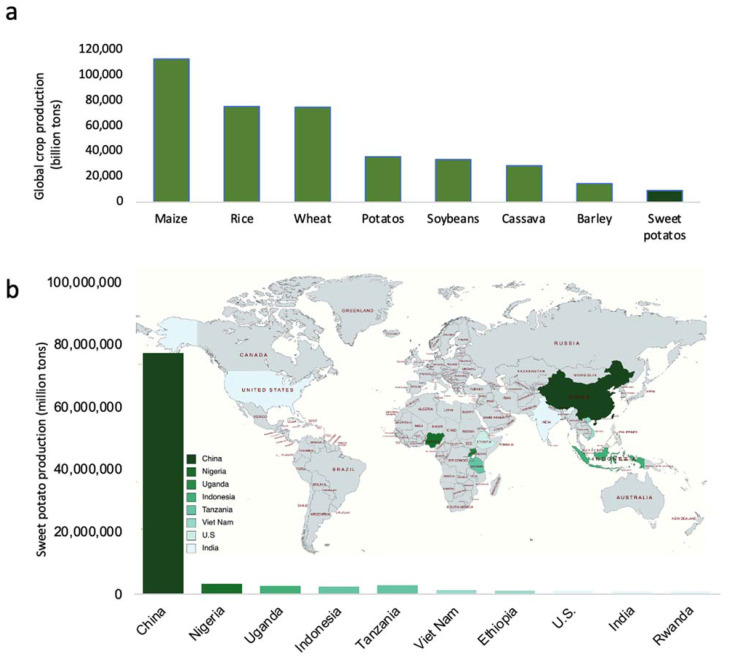
Global production of major crops including sweet potato. (**a**) The production of the top eight major crops in the world; (**b**) the global distribution and the top ten sweet potato-producing countries in 2018. The darkness of the color reflects the production volumes. The production numbers were generated according to the Food and Agriculture Organization (FAO) Statistics 2019.

**Figure 2 ijms-22-09533-f002:**
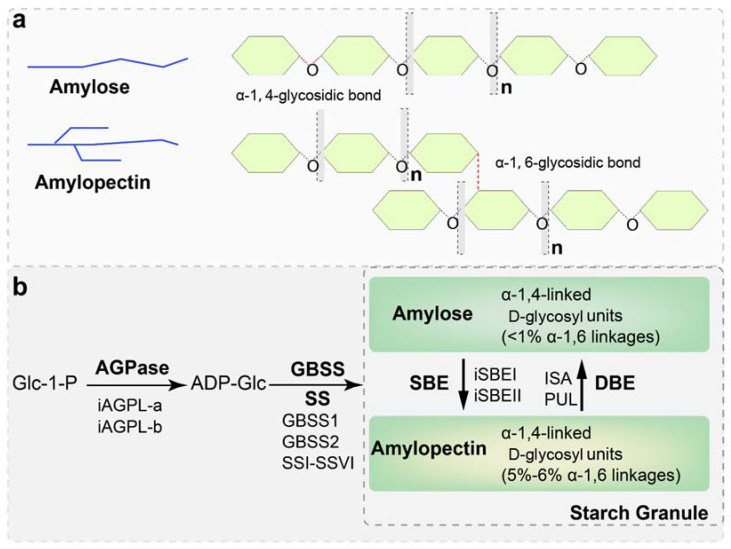
Schematic presentation of amylose and amylopectin structure (**a**) and core enzymes for starch biosynthesis (**b**). There are two major glycosidic bonds in the starch molecules, α-1,4-glycosidic bond and α-1,6-glycosidic bond. The branched amylopectin chain is built from mostly short α-1,4-glucan chains (as present in amylose) linked by α-1,6-glycosidic branching points (**a**). Starch biosynthesis in sweet potato storage roots requires a multitude of enzyme activities (**b**). Starch biosynthesis starts with the conversion of sugar adenosine diphosphate glucose (ADP-Glu) from glucose 1-phosphate (Glu-1-P), catalyzed by ADP-glucose pyrophosphorylase (AGPase). ADP-Glc is the precursor of amylose and amylopectin biosynthesis. Granule-bound starch synthase (GBSS) elongates the linear α-(1,4)-glucan chains by adding a glucose unit from ADP-Glc to the non-reducing end, whereas the soluble starch synthase (SS) and starch branching enzyme (SBE) catalyze amylopectin production. Debranching enzymes, isoamylase (ISA), and pullulanase (PUL) maintain the correct assembly of the starch granule by efficiently hydrolyzing the α-(1,6)-linkage in amylopectin. The activities between the branching and debranching enzymes determine the quality, size, and shape of starch granules.

**Figure 3 ijms-22-09533-f003:**
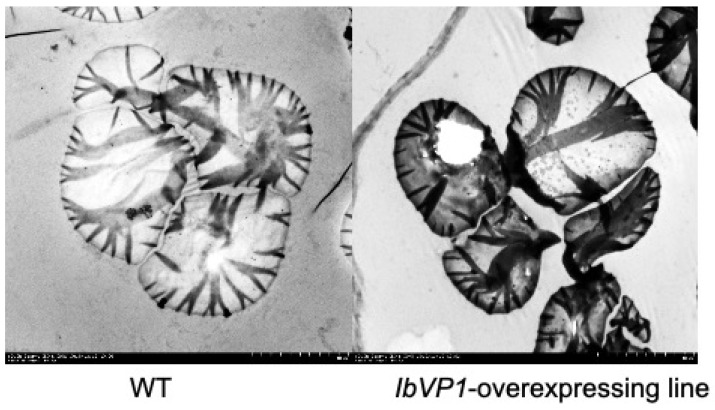
Ultrastructure of starch granules from wild-type and engineered sweet potato. Starch granules were examined by a transmission electron microscope (TEM). In wild-type plants, the starch granules show typical morphology of “zebra stripes” that are crystalline lamella, which may be associated with double helical amylopectin [84]. The granules and the stripes are darker and thicker in starch from the IbVP1-overexpressing lines. In addition, the hilum cracks (centric holes) in the starch granules of the transgenic line become more dominant, likely affecting the water absorption and swelling of the starch. The cracking on hilum is also observed when downregulating SSII. IbVP1, H + -pyrophosphatase. Scale bar, 5 μm.

**Figure 4 ijms-22-09533-f004:**
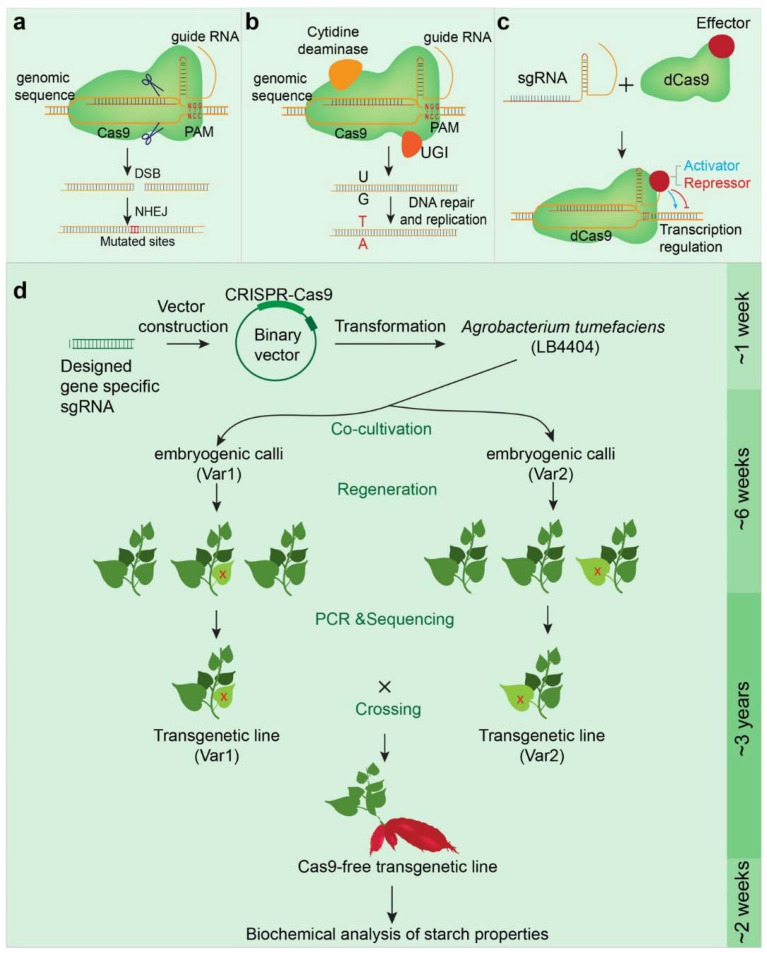
The multifunctional CRISPR/Cas9 system and a schematic description of sweet potato genome-editing and selection for T-DNA-free progeny. (**a**) The basic CRISPR/Cas9 system. The designed guide RNA, based on a specific target DNA sequence, binds to the target site, and directs the Cas9 protein to the genomic sequence complementary to sgRNA, adjacent to a protospacer adjacent motif (PAM). PAM comprises three nucleotides “NGG”, where *n* represents any nucleotide. Cas9 acts as the endonuclease that cuts the DNA sequence specifically recognized by the sgRNA. Upon cleavage, the genomic DNA forms double-strand breaks (DSBs). The DSBs are then repaired by non-homologous end joining (NHEJ), a conserved mechanism that repairs the DSB in the genome. (**b**) Base editing technology. Cytidine deaminase is fused to an inactivated Cas9 protein to generate a Cas9 nickase (nCas9) that acts as a cytosine base editor (CBE). CBE generates C·G to T·A base substitutions. UGI is the uracil DNA glycosylase inhibitor. The important function of UGI is to prevent mutagenesis by eliminating uracil from DNA molecules by cleaving the N-glycosidic bond and initiating the base-excision repair pathway. To generate A·T to G·C substitutions, the cytidine deaminase is replaced by an adenine deaminase to create an adenine base editor (ABE). (**c**) The CRISPR interference (CRISPRi). The inactivated dCas9 (endonuclease deficient Cas9), which lacks endonuclease activity but retains guide RNA binding activity, is fused with transcriptional effectors (activators or repressors). Upon binding to the promoter of a target gene, guided by a designed guide RNA, the effector activates (by an activator fusion) or represses (by a repressor fusion) the target gene expression without changing the DNA sequence. (**d**) A working flow illustrating genome editing and selection of T-DNA-free progeny of tuber crop plants, such as sweet potato [16], that are normally bred through tissue propagation rather than seed selection. In short, a binary vector (pCambia 1300) containing CRISPR/Cas and sgRNA is used to transform sweet potato mediated by *Agrobacterium tumefaciens* (LB4404). Due to the self-incompatibility of sweet potato, two varieties (Var1 and Var2) were transformed to generate calli. The regenerated seedlings were selected by PCR and DNA sequencing to identify transgenic lines (represented by dark green leaves) that are mutated at the target sites (red X). The two variety transgenic lines were then crossed to generate segregating F1 hybrids that contain the mutation, but with or without the T-DNA (Cas9 and guide RNA). Only the lines with the mutation but without T-DNA (-Cas9) are selected and moved forward for subsequent characterization. CRISPR/Cas has shown significant applicability in genome editing-based crop breeding and will have a profound impact on the future of agriculture [33,120]. CRISPR technology accelerates the process of plant breeding and is continuously evolving. The base editors and primer editors, recently developed on the CRIPSR/Cas platform, enable precise mutations of single nucleotides, or deletions and insertions of DNA fragments in a genomic position [121,122]. Gene editing is also advantageous when multiple genes must be mutated. For instance, the inactivation of both *SBEIIa* and *SBEIIb* in wheat reduced amylose by more than 70%. However, no changes in amylose content is detected when only *SBEIIa* or *SBEIIb* is mutated [123]. Simultaneously mutating multiple genes can be achieved by CRISPR/Cas [124].

**Table 1 ijms-22-09533-t001:** Biotechnological improvement of starch quality traits in sweet potato.

*Gene*	Technique	Cultivar	Improved Trait/Observation	Reference
*GBSSI*	Co-suppression; RNAi	Kokei-14	Reduction of amylose content	[18]
			Tuberous roots free of amylose generated in transgenic plants, compared to 17–18% in wild-type plants (WT).	
			Amylose-free starch obtained from transgenic plants.	[19]
			Decreased content of DP 6–8 amylopectin.	
			The amylose-free starch exhibits altered values of gelatinization properties, with increased T_o_ at 71 °C, T_p_ at 74 °C, and ΔH at 17.2 J/g.	
			Irregular pasting curve was observed in amylose-free starch	
			Amylose-free transgenic plants	[20]
			No significant difference in storage root yield between transgenic and non-transgenic plants	
			Amylose-free; lacking long chain amylopectin (DP > 100) and decreased DP 6–7 amylopectin in transgenic plants	[21]
	RNAi	Xushu-22 (High-starch)	Amylose content in transgenic (waxy) lines decreased to 5.7% from 30.4% (Xushu22)	[22]
			Larger granule size was observed in waxy starch from transgenic plants.	
			- Fewer short chain of amylopectin CLDs.	
			The waxy transgenic plants show weak Type II distribution compared to high amylose starch	[23]
			Large decrease in amylose and increase in long chain amylopectin.	
	CRISPR/Cas9	Xushu-22	Lower amylose content (5.8–22.4%) in mutant lines compared to WT (27.2%)	[16]
		Taizhong-6 (Carotenoid rich)	Lower amylose content (5.5–14.8%) in mutant lines compared to WT (25.7%)	
*SSI*	Overexpression and RNAi	Lizixiang (Low starch)	Increased starch content and granule size and reduced the proportion of amylose in *IbSSI* overexpressing plants. Starch content reduced in RNAi plants.	[7]
*SSII*	RNAi	Quick Sweet	Lower pasting temperature (Pte) and breakdown (BD) of starch from transgenic plants compared to WT.	[17]
			Higher amylopectin at DP 6–11 but lower at DP 13–25.	
			Cracking on the hilum was observed in starch granules from mutants.	
*SBEII*	RNAi	Kokei-14	Higher in amylose content but slightly lower in starch yield, and showed different shapes of starch grains compared to WT.	[24]
			Increased levels of amylose, phosphate contents, amylopectin at DP12–15 and DP24–33, and altered crystalline structure. Decreased amylopectin at DP >100, DP6–11, compared to WT.	[25]
	CRISPR/Cas9	Xushu-22	Higher amylose content (38.0–40.3%) compared to WT (27.2%)	[16]
			Fewer short chain amylopectin (DP6–12)	
		Taizhong-6	Higher amylose content (37.4–37.8%) compared to WT (25.7%)	[16]
*SBEI* *SBEII*	RNAi	Xushu-22	High amylose content in transgenic plants (50.3–65.5%).	[22]
			Larger granule size in high-amylose starch (90 µm) compared to WT (5–60 µm).	
			Less short chain amylopectin compared to WT.	
			Higher PTe (85–89 °C) compared to WT (80 °C).	
			Lower BD (58.0–75.7 cP) and higher setback (SB) (911–1810 cP) compared to WT (BD = 403 cP, SB = 425 cP).	
			Lower gelatinization level and higher retrogradation ability compared to WT.	
			Increased Type II semicrystalline lamellae.	[23]
			Chain length distributions (CLDs) of the high-amylose composes of three fractions, DP 32, DP 33–200, and the combination of first and second fractions.	

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
