# Peer review of "Engineering Properties of Sweet Potato Starch for Industrial Applications by Biotechnological Techniques including Genome Editing"

_ijms, 2021, doi:10.3390/ijms22179533_

Round 1

Reviewer 1 Report

Dear Authors,

Line 33:  «Sweet potato (Ipomoea batatas) is the seventh-largest food crop in the world (Figure 1).»

In fact, Figure 1 does not show sweet potato is the 7th crop. It says, that China produces majority of it in the world.

Line 48: It is better to compare sweet potato with potato, not with cereals, because sweet potato and potato starch are both tuber starches. For review of potato starch biosynthesis and related genes\proteins refer to Khlestkin V.K., Peltek S.E., Kolchanov N.A. Target genes for development of potato (Solanum tuberosum L.) cultivars with desired starch properties (review) DOI: 10.15389/agrobiology.2017.1.25rus.

Line 76 and 230: “different industrial applications“ – Comprehensive review about various starch applications and properties has to be referred: Review of direct chemical and biochemical transformations of starch Vadim K. Khlestkin, Sergey E. Peltek, Nikolay A. Kolchanov DOI 10.1016/j.carbpol.2017.10.035.

Line 184: “The transgenic waxy sweet potato with high amylose have larger starch granule sizes 184 (70 and 90 μm) compared with the wild-type (WT) [48]. Reduction in amylose content reduces the average diameter of the starch granules; hence, high amylose content tends to increase the granule size [48].” Not clear, if high amylose results into larger granules, or smaller? The same reference is given as confirmation for both.

Line 208: “The starch from sweet potato (with average chain length of 20.4-24.7) and potato (20.5) have a similar range of CLD values, which are higher than those of cassava (19.3) and maize (18.9) [54].” -  Not clear which values are compared. It is better to provide the values for comparison.

Some polishing of English language (use of articles, prepositions, etc) required

Author Response

Point 1: Line 33:  Sweet potato (Ipomoea batatas) is the seventh-largest food crop in the world (Figure 1). In fact, Figure 1 does not show sweet potato is the 7th crop. It says, that China produces the majority of it in the world.

Response 1: We apologize.  We changed Figure 1 to include the global production of major crops. We also changed the description of sweet potato as “one of the major crops”.

Point 2: Line 48: It is better to compare sweet potato with potato, not with cereals because the sweet potato and potato starch are both tuber starches. For a review of potato starch biosynthesis and related genes\proteins refer to Khlestkin V.K., Peltek S.E., Kolchanov N.A. Target genes for development of potato (Solanum tuberosum L.) cultivars with desired starch properties (review) DOI: 10.15389/agrobiology.2017.1.25rus.

Response 2: Thank you. In the revised manuscript, we made the comparison of sweet potato with potato. We cited the suggested reference “DOI: 10.15389/agrobiology.2017.1.25rus ”, even though it is not published in English.

Point 3: Line 76 and 230: “different industrial applications“ – Comprehensive review about various starch applications and properties has to be referred: Review of direct chemical and biochemical transformations of starch Vadim K. Khlestkin, Sergey E. Peltek, Nikolay A. Kolchanov DOI 10.1016/j.carbpol.2017.10.035.

Response 3: We included this nice paper as suggested.

Point 4: Line 184: “The transgenic waxy sweet potato with high amylose have larger starch granule sizes 184 (70 and 90 μm) compared with the wild-type (WT) [48]. Reduction in amylose content reduces the average diameter of the starch granules; hence, high amylose content tends to increase the granule size [48].” Not clear, if high amylose results into larger granules, or smaller? The same reference is given as confirmation for both.

Response 4: Thank you. We have modified it accordingly. “Reduction in amylose content reduces the average diameter of starch granules, in other words, higher amylose content tends to produce larger granules [10, 58].”

Point 5: Line 208: “The starch from sweet potato (with average chain length of 20.4-24.7) and potato (20.5) have a similar range of CLD values, which are higher than those of cassava (19.3) and maize (18.9) [54].” -  Not clear which values are compared. It is better to provide the values for comparison.

Response 5: We changed the sentence to “The starch from sweet potato (with average chain length (ACL 20.4-24.7) and potato (VCL 20.5) have a similar range of CLD values, which are higher than those of cassava (VCL 19.3) and maize (ACL 18.9)”.

Point 6: Some polishing of English language (use of articles, prepositions, etc) required.

Response 6: We made additional efforts to polish the language.

Reviewer 2 Report

Review Comments

This review article discusses the recent development in sweet potato breeding, understanding of starch properties, and the progress in sweet potato genomics. It describes the applicational values of sweet potato starch in food, industrial products, and biofuel, as well as the effects of starch property in different industrial applications. It also explores the possibility of manipulating starch properties through biotechnological means, such as the CRISPR/Cas-based genome editing. The ability to target the genome with precision opens new doors of opportunity for reducing breeding time, increasing yield, and optimizing the starch property of sweet potatoes.

- The review is well-written. However, some improvements and revisions are still required as shown below;

- There many points/sections (section 2 and 3, in particular) need more discussion in the view of the present literature. Recent literature on these topics needs to be added too.

- The references section should include more studies about the recent literature presented on this topic as well.

- The discussion section is VERY SHORT and should be extended to discuss the views and needs of this article.

- The article lacks a conclusion section. A conclusion section needs to be added to conclude all the most important information of this article and future perspectives.

- English language of the article needs improvements and corrections by native English speaker or English proofreading service.

Author Response

This review article discusses the recent development in sweet potato breeding, understanding of starch properties, and the progress in sweet potato genomics. It describes the applicational values of sweet potato starch in food, industrial products, and biofuel, as well as the effects of starch property in different industrial applications. It also explores the possibility of manipulating starch properties through biotechnological means, such as the CRISPR/Cas-based genome editing. The ability to target the genome with precision opens new doors of opportunity for reducing breeding time, increasing yield, and optimizing the starch property of sweet potatoes.

Point 1: - The review is well-written. However, some improvements and revisions are still required as shown below.

- There many points/sections (section 2 and 3, in particular) need more discussion in the view of the present literature. Recent literature on these topics needs to be added too.

Response 1: Thank you for your suggestions. We extended the section 2 and 3. In Section 2, we provided more details about starch metabolism and recent progress. In Section 3, we explained the tight connections among genes, starch metabolism, starch properties, and starch products. We also included starch phosphorylation in Section 3 as starch phosphorylation is a hot topic in starch research. We argued that manipulation of starch properties through genetic engineering is advantageous compared to industrial treatment.

Point2: The references section should include more studies about the recent literature presented on this topic as well.

Response 2: We added more recent publications, especially those published in past two years.

Point3: - The discussion section is VERY SHORT and should be extended to discuss the views and needs of this article.

Response 3: We expanded the discussion and added a Conclusion.

Point4:- The article lacks a conclusion section. A conclusion section needs to be added to conclude all the most important information of this article and future perspectives.

Response 4: We added a conclusion section.

Point5:- English language of the article needs improvements and corrections by native English speaker or English proofreading service.

Response 5: We have made efforts to polish the language.

Reviewer 3 Report

Dear Authors,

The manuscript submitted for review appears to be a conglomeration of rather random sections that are in no way related to the title. Section 5 mainly describes the genome editing method. This is not a new technique that requires explaining its basics to readers. However, it does not provide information on what has been achieved so far in genome editing in this species. Information on starch biosynthesis can be found in both the introduction and Section 2. Which makes it significantly more difficult to read. Overall, I feel that the manuscript does not provide any added value over previous publications in the field.

Best regards,

M.

Author Response

Point 1: “The manuscript submitted for review appears to be a conglomeration of rather random sections that are in no way related to the title. Section 5 mainly describes the genome editing method. This is not a new technique that requires explaining its basics to readers. However, it does not provide information on what has been achieved so far in genome editing in this species. Information on starch biosynthesis can be found in both the introduction and Section 2. Which makes it significantly more difficult to read. Overall, I feel that the manuscript does not provide any added value over previous publications in the field.”

Response 1: We apricated your comments and made a significant revision of the manuscript.

We feel that, in order to help reader, appreciate the title subject, which is summarized in Table 1, much background information needs to be provided. In the revised manuscript, significant changes were made to section 2 and section 3. Section 2 focuses on the starch biosynthetic pathway, whereas section 3 describes how gene regulation affects starch functionality. Overlapping information related to starch biosynthesis was removed from the Introduction. We also made significant revisions to section 5 (CRISPR). We reduced descriptions of the technique and other technologies that are unrelated to CRISPR. We expanded the Discussion and provided a Conclusion. We aimed to convey to readers that biotechnological approaches to modify starch properties are sound and effective and that the new advancement in genome editing of sweet potato further expands our ability to develop novel germplasms with desired starch properties.

Round 2

Reviewer 1 Report

Dear Authors,

I have no more corrections. Thank you for your job done.

Reviewer 2 Report

The revised version is greatly improved as per my suggested comments 

Reviewer 3 Report

As it stands, the manuscript is definitely better.